# Reelin Regulates Developmental Desynchronization Transition of Neocortical Network Activity

**DOI:** 10.3390/biom14050593

**Published:** 2024-05-17

**Authors:** Mohammad I. K. Hamad, Obada Rabaya, Abdalrahim Jbara, Solieman Daoud, Petya Petrova, Bassam R. Ali, Mohammed Z. Allouh, Joachim Herz, Eckart Förster

**Affiliations:** 1Department of Anatomy, College of Medicine and Health Sciences, United Arab Emirates University, Al Ain 17666, United Arab Emirates; m_allouh@uaeu.ac.ae; 2Department of Neuroanatomy and Molecular Brain Research, Medical Faculty, Ruhr University Bochum, 44801 Bochum, Germany; obada.rabayah@gmail.com (O.R.); soulieman-daoud@hotmail.com (S.D.); petya.petrova@rub.de (P.P.); eckart.foerster@rub.de (E.F.); 3Department of Genetics and Genomics, College of Medicine and Health Sciences, United Arab Emirates University, Al Ain 17666, United Arab Emirates; bassam.ali@uaeu.ac.ae; 4Departments of Molecular Genetics, Neuroscience, Neurology and Neurotherapeutics, Center for Translational Neurodegeneration Research, University of Texas Southwestern Medical Center, Dallas, TX 5323, USA

**Keywords:** reelin, developmental desynchronization, network activity, GABA receptors, synchronized network activity, neocortex

## Abstract

During the first and second stages of postnatal development, neocortical neurons exhibit a wide range of spontaneous synchronous activity (SSA). Towards the end of the second postnatal week, the SSA is replaced by a more sparse and desynchronized firing pattern. The developmental desynchronization of neocortical spontaneous neuronal activity is thought to be intrinsically generated, since sensory deprivation from the periphery does not affect the time course of this transition. The extracellular protein reelin controls various aspects of neuronal development through multimodular signaling. However, so far it is unclear whether reelin contributes to the developmental desynchronization transition of neocortical neurons. The present study aims to investigate the role of reelin in postnatal cortical developmental desynchronization using a conditional reelin knockout (Reln^cKO^) mouse model. Conditional reelin deficiency was induced during early postnatal development, and Ca^2+^ recordings were conducted from organotypic cultures (OTCs) of the somatosensory cortex. Our results show that both wild type (wt) and Reln^cKO^ exhibited an SSA pattern during the early postnatal week. However, at the end of the second postnatal week, wt OTCs underwent a transition to a desynchronized network activity pattern, while Reln^cKO^ activity remained synchronous. This changing activity pattern suggests that reelin is involved in regulating the developmental desynchronization of cortical neuronal network activity. Moreover, the developmental desynchronization impairment observed in Reln^cKO^ was rescued when Reln^cKO^ OTCs were co-cultured with wt OTCs. Finally, we show that the developmental transition to a desynchronized state at the end of the second postnatal week is not dependent on glutamatergic signaling. Instead, the transition is dependent on GABA_A_R and GABA_B_R signaling. The results suggest that reelin controls developmental desynchronization through GABA_A_R and GABA_B_R signaling.

## 1. Introduction 

During early neocortical development, neurons exhibit a high SSA pattern which manifests as synchronized oscillations in intracellular Ca^2+^ concentration [1]. These spontaneous Ca^2+^ waves usually start in the posterior cortex, propagate slowly towards its anterior end [2], and include multiple areas within the central nervous system such as the retina, spinal cord, hindbrain, hippocampus, and neocortex [3]. Moreover, it has been observed in a variety of species [4]. Moreover, neocortical SSA strongly depends on AMPA and NMDA receptor activation and is modulated by cholinergic activity in the first postnatal week [2,5,6]. In the absence of sensory stimuli, cortical networks exhibit complex, self-organized activity patterns [7,8,9]. Extracellular multielectrode recordings were used to observe three distinct patterns of synchronized oscillatory activity in the rat somatosensory cortex in vivo at postnatal day (P) 0 to 7. Among these, the spindle bursts, which are short bursts of 10 Hz oscillations, were found to be the dominant activity pattern in S1 [10], confirming previous findings [11,12]. In vivo, the intact somatosensory cortex of newborn rats is spatially confined to spindle bursts, which represent the first and only organized network pattern. These spindles are selectively triggered in a somatotopic manner by spontaneous muscle twitches, which are motor patterns analogous to human fetal movements [11]. Early neuronal activity also controls the rate of programmed cell death in the developing barrel cortex, suggesting that spindle bursts are physiological activity patterns particularly suited to suppressing apoptosis [13]. Moreover, the SSA is a hallmark of neuronal circuits during early development [14]. For instance, neurons in the developing auditory system exhibit spontaneous bursts of activity before hearing onset. When glutamate release from hair cells is blocked, the changes in peripheral excitability maintain patterned neural activity in the brain [15]. This early form of activity is typically discontinuous and interrupted by silent periods, as reported in studies on humans [16], in the rat visual system [17], and in the visual cortex of preterm human infants as well as neonatal rats [18]. During the initial days of postnatal development, the SSA relies on gap junctions for its functioning [12,19], whereas it helps in spreading this wave-like activity in the second postnatal week [9].

At the end of the second postnatal week, neocortical networks undergo a transition to a desynchronized sparse state that lacks a clear spatial structure [7]. This has also been confirmed in studies with mice in vivo at P13 [20], in rats between P25 and 36 [21], in rats between P21 and 30 [22], and in mice at P14 [23]. The transition to a desynchronized sparse state has been suggested to be a mechanism that allows for increased storage capacity in associative memories while saving energy at the same time [24]. Additionally, there has been controversy regarding whether the developmental desynchronization transition depends on synaptic input. For instance, the deprivation of sensory input from the periphery had no effect on the time course of the developmental transition in the neocortex [7]. On the other hand, in the absence of coincident synaptic input in the cerebellum, Golgi cells synchronize their firing. In turn, with coincident mossy fiber input in the cerebellum, a desynchronization transition is triggered [25]. 

During early development, GABA has been suggested to play a role in controlling neuronal differentiation, synaptogenesis, and synaptic plasticity through its excitatory effects [1,26]. The role of GABA in regulating SSA in the hippocampus and neocortex during early development has been extensively investigated over the past two decades [27]. In the early postnatal week of hippocampal development, the SSA of living mice is characterized by sharp waves named giant depolarizing potentials (GDPs) [28], which depend on excitatory GABA [29,30]. The release of GABA in early postnatal neocortical development induces Ca^2+^ influx; however, it diminishes neuronal firing [31,32]. This suggests that although GABA is depolarizing, it predominantly inhibits network activity through shunting. In comparison to the hippocampal GDPs, GABA is not involved in neocortical SSA in the first postnatal developmental week, as confirmed by other studies [31,33,34,35]. It is important to note that the developmental shift from GABA excitation to inhibition occurs simultaneously with the shift towards network desynchronization at the end of the second postnatal week. This suggests that inhibitory GABA action is necessary for the network desynchronization shift.

Reelin is an extracellular protein which is involved in mammalian brain development and function. It controls neuronal migration and layer formation in the developing cerebral cortex, as evidenced by morphological studies on reeler mutant mice that lack reelin expression [36,37]. Reelin also modulates synaptic plasticity in the adult brain [38,39]. It has been shown that in the absence of reelin, experience-dependent plasticity in the visual cortex into adulthood was prolonged [40]. The confirmation of the contribution of reelin signaling to the genetic and behavioral expression of learning and memory has been documented [41,42]. Moreover, an epigenomic analysis has revealed that the binding of reelin to its receptor results in the expression of certain immediate early genes involved in synaptic plasticity [43]. Reelin signaling, mediated via the cell membrane located reelin receptors ApoER2 and VLDLR, guides the radial migration of newborn neurons and orchestrates the development of cortical layers [44,45,46]. In the developing and adult brain, the majority of GABAergic interneurons in the neocortex expresses reelin. Additionally, reelin exerts opposing effects on dendritic growth depending on the developmental stage. During the embryonic developmental stage, it enhances dendritic growth [47,48,49,50], whereas it limits dendritic growth during postnatal development [50,51,52,53]. Adult conditionally induced Reln^cKO^ mice exhibited altered hippocampal LTP and a subtle behavioral phenotype. However, their cortical architecture was indistinguishable from that of their wild-type littermates [54]. The question of whether reelin deficiency affects the early cortical network transition from synchronized to a desynchronized state remains to be answered. To address this question, we examined how conditionally induced reelin deficiency after birth affects developmental network desynchronization in the somatosensory neocortex of Reln^cKO^ mice. We focused on the potential interaction between reelin signaling and GABA_B_R function in regulating the developmental desynchronization transition, as synaptic release is primarily controlled by GABA_B_Rs that mediate inhibitory GABA effects at the end of the second postnatal week.

## 2. Material and Methods 

### 2.1. Reelin Conditional Knockout Mice (Reln^cKO^) 

The mice were kept in a standard 12 h light cycle and provided with standard mouse chow ad libitum. The care and use of experimental animals were in compliance with the Federal German law and ARRIVE guidelines, with permission Nr. 84-02.04.2016.A383. The conditional Reln^cKO^ line was generated as previously described [54]. Conditional reelin knockout mice (Reln^flox/flox^ CAG-^CreERT2^ mice) were obtained by crossing Reln^flox/flox^ mice with hemizygous tamoxifen-inducible Cre-recombinase-expressing mice (CAG^CreERT2^) [55]. For the experiments, we selected only male mice with Reln^flox/flox^ CAG^CreERT2^ and crossed them with female mice with Reln^flox/flox^ to generate Reln^flox/flox^ wildtype (wt) and Reln^flox/flox^ CAG^CreERT2^ (Reln^cKO^) siblings, as verified by PCR. This study was not pre-registered, and no exclusion criteria were pre-determined, making it an exploratory study. We did not perform blinding or randomization to allocate subjects in the study, and no sample size calculation was performed.

### 2.2. PCR and Genotyping

DNA was isolated from ear, tail, and brain tissue samples using a ReliaPrep gDNA kit (Cat# A205, Promega, Mannheim, Germany). The isolated DNA amounts were determined by spectrophotometry using the Genova Nano system (Jenway). PCR reactions were performed in a 50 μL reaction mixture containing 200 ng of template DNA, Soriano buffer (0.67 M Tris, 0.16 M ammonium sulfate, 67 mM MgCl2, 67 µM EDTA, and 50 mM β-Mercaptoethanol), Taq polymerase, 2 µL DMSO, and 10 mM dNTPs. The following primers were used for genotyping wildtype mice: forward primer 5′-ATAAACTGGTGCTTATGTGACAGG-3′, reverse primer 5′-AGACAATGCTAACAACAGCAAGC-3′ (450 bp); for Reln^flox/flox^ mice: forward primer 5′-GCTCTGGCCAAGCTTTATC-3′, reverse primer 5′- CGCGATCGATAACTTCGTATAGCATAC-3′ (1200 bp); and for the detection of CAG-CreERT2: forward primer 5′-ATTGCTGTCACTTGGTCGTGG-3′, reverse primer 5′-GGAAAATGCTTCTGTCCGTTTGC-3′ (200 bp). The PCR amplification products were verified on a 2% agarose gel in TBE buffer.

### 2.3. Organotypic Cultures and Pharmacological Treatment

OTCs were prepared from newborn mice on postnatal day 0 (P0). All solutions used for OTC preparation were sterile. OTC preparation was performed in a laminar airflow bench with horizontal counterflow (Horizontal Flow, ICN, Biomedicals, Eschwege; Germany). Mice were briefly anesthetized with hypothermia and decapitated. The skull was gently removed, and the cortex was placed on a chopper plate (McIllwain, Bad Schwalbach, Germany). The somatosensory cortex was cut into 350 μm thick slices, and the slices were gently transferred into ice-cold buffered salt solution (GBSS, Cat# 24020117, Gibco, Eggenstein, Germany) containing 25 mM D-glucose. After recovery for 30 min, the sections were transferred to coverslips (12 × 24 mm, Kindler). A mixture of chicken plasma (Sigma) and GBBS/thrombin (Merck) was mixed in a ratio of 2:1 and allowed to clot for 45 min. The coverslip was transferred to a roller tube (Nunc) filled with 750 μL of semi-artificial medium and placed in a roller incubator at 37 °C. For knockout induction, OTCs after preparation were stimulated with 1 µM (Z)-4-hydroxytamoxifen (4-OHT) (Cat# 3412, Tocris, Wiesbaden, Germany) at 2 days in vitro (2 DIV) for 5 consecutive days and maintained for experiments until 14 DIV. A graphical flow chart of the OTC experimental procedures is shown in Figure 1A. The following drugs were used for Ca^2+^ imaging experiments: Gabazin (10 µM, Cat# 0130, Tocris, Wiesbaden, Germany), CGP35348 (10 µM, Cat# 1245/10, Tocris, Wiesbaden, Germany), CNQX (10 µM, Cat# 0190/10, Tocris, Wiesbaden, Germany), APV (50 µM, 0190/10, Tocris), recombinant mouse LRPAP protein (LDL receptor-related protein-associated protein 1 (also called RAP), 50 ng/mL, Cat# 4480-LR, R&D Systems).

### 2.4. Ca^2+^ Imaging Using Spinning Disc Laser Microscopy

Ca^2+^ imaging was performed with the Ca^2+^ indicator Oregon Green 488 BAPTA-1 AM (OGB-1 AM, Cat# O6807, Molecular Probes, Eugene, OR, USA) for OTC loading at DIV14, as we described previously [56,57]. Briefly, a solution of 20% PF127 (Cat# P2443, Sigma, Steinheim, Germany) dissolved in DMSO (*w*/*v*) (J.T. Baker) was prepared to dissolve OGB-1 (1 µM). After loading, the sections were washed several times to remove excess dye and allowed to recover for one hour. The slices were transferred to the recording chamber mounted on the fixed stage of an inverted microscope and perfused with 95% O_2_, 5% CO_2_ bubbled artificial cerebrospinal fluid (ACSF; 3–5 mL/min) at 32 ± 2 °C. The composition of the ACSF was as follows: 125 mM NaCl, 5 mM KCl, 2 mM CaCl_2_, 1 mM MgSO_4_, 25 mM NaHCO_3_, 1.25 mM NaH_2_PO_4_, 25 mM glucose, pH 7.4. Osmolality was 295 ± 5 mOsm, as determined by a cryoscopic osmometer (Osmomat 030, Gonotec, Berlin, Germany). Fluorometric Ca^2+^ recordings were performed with a Visiscope spinning-disc confocal system CSU-W1 (Visitron, Munich, Germany) consisting of a CSU-W1-T2 spinning disc unit and an sCMOS digital camera (4.2 Mpixel rolling shutter version) on an inverted Nikon Ti-E motorized microscope using a CFI P-Fluor 20× objective (NA 0.5, WD = 2.10 mm). Images were acquired at 3 frames per second with an exposure time of 330 ms using VisiView image acquisition software (Visitron, Munich, Germany). The Ca^2+^ dye OGB-1 was excited at 488 nm. The emitted fluorescence was collected through an ET 525/50 filter. 

### 2.5. Analysis of Imaging Data

From each animal, we imaged 3–4 OTCs from the somatosensory cortex. In each OTC, 3 areas of interest (AOIs) were recorded. Fluorometric data are expressed as ΔF/F^0^ (background-corrected fluorescence increase divided by resting fluorescence). Raw data, provided as an 8-bit linear intensity scale, were plotted as fluorescence intensity versus time. After background correction, the cell soma was selected as a region of interest (ROI). Baseline fluorescence (F^0^) was calculated as the average of 20 frames in a time window without neuronal activity (judged by visual inspection). Data were then normalized to mean fluorescence intensities [ΔF/F^0^ = (F − F^0^)/F^0^] to allow comparison of data across experiments. Recorded neurons with ΔF/F^0^ ≥ 10% were considered active and included in the analyses. The result was manually inspected and corrected as necessary. Fractions of synchronized neurons were calculated as follows: the number of neurons exhibiting a simultaneous Ca^2+^ event, having ΔF/F^0^ ≥ 10%, and participating in at least 40% of the network events during the trace was divided by the total number of active neurons. Similarly, the fractions of desynchronized neurons were calculated as follows: the remaining neurons that exhibited a Ca^2+^ event and had ΔF/F^0^ ≥ 10% but did not participate in any of the synchronous network events during the trace were divided by the total number of active neurons. Neurons with ΔF/F^0^ < 10% were considered silent. 

### 2.6. Statistical Analysis 

The statistical analyses were performed using Sigma Stat 12 (SPSS Incorporated). For comparisons between two groups, Student’s unpaired t-test was used when the normality test (Shapiro–Wilk) passed; otherwise, the Mann–Whitney test was used. For comparisons between more than two groups, one-way ANOVA was used, followed by a Holm–Sidak multiple comparison test for post hoc analysis if the normality test was passed. If normality assumptions were not met, we conducted a one-way ANOVA on ranks, followed by Tukey’s multiple comparison test for post hoc analysis, which was used to identify significant groups. Statistical significance was set at *p* < 0.001.

## 3. Results 

### 3.1. Developmental Desynchronization of Cortical Network Activity in the Somatosensory Cortex

Eliminating reelin during embryonic development affects cortical layer lamination [58]. However, after birth, when neurons have already migrated and settled into their proper layers, elimination of reelin does not affect cortical lamination. In a previous study, we found that conditional elimination of reelin after birth did not alter the distribution of layer-specific expression markers in the neocortex, thus preserving cortical layers [59]. To bypass the cortical layer malformations present in the reeler mutant, 4-OHT was added to the culture medium of OTCs from 2 DIV for 4 consecutive days to induce reelin deficiency. To investigate developmental network desynchronization in the mouse somatosensory cortex, we recorded Ca^2+^ activity in spontaneously active OTC at different developmental time windows: 5, 10, and 14 DIV (Figure 1). Imaging large networks of neuronal populations loaded with the calcium indicator OGB-1 using a spinning-disk confocal microscope equipped with fast cameras, can be used to image thousands of neurons simultaneously without significant photobleaching, with good signal-to-noise ratio, and minimal cellular damage [56,60]. This technique does not require electrodes penetrating the tissue, minimizing cellular damage. Ca^2+^ recording of OTCs loaded with the calcium indicator OGB-1 showed a synchronous neuronal activity pattern in the wt OTC at 5 DIV (Figure 1B–D). Around 80% of cortical neurons displayed synchronous neuronal activity at that stage, 5% of the neurons showed desynchronized activity, and the remaining 15% were silent. At 10 DIV, there was an increase in the percentage of neurons exhibiting desynchronized activity (60%, Figure 1E–G). By 14 DIV, 90% of recorded cortical neurons of wt cultures exhibited desynchronized activity, making it the dominant form of neuronal activity during this developmental time window (Figure 1H–J). The observed developmental desynchronization pattern in the somatosensory cortex in this study was comparable to neuronal activity patterns which were previously reported in in vivo studies. Similar patterns were reported in mice by [20] around P13 and by [23] at P14, while similar patterns in rats were reported by [21] between P25 and 36 and by [22] between P21 and 30.

### 3.2. Developmental Network Desynchronization Is Disrupted in Reelin-Deficient Mice

During embryonic and early postnatal development, Cajal–Retzius cells express reelin. However, these cells undergo selective cell death through apoptosis and disappear completely by P14 [46,61,62]. Despite this, a subset of GABAergic inhibitory interneurons in the cortex [63,64,65,66] and entorhinal cortical ocean cells [67] continue to express reelin postnatally. The function of reelin in regulating neuronal activity during early cortical development remains poorly understood. Previous studies have shown that reelin regulates the function of GABA_B_Rs [59] and dendritic development [53]. Initially, we aimed to confirm that 4-OHT treatment induces nuclear Cre activation and knockout of the floxed reelin gene. Therefore, we performed immunofluorescence at different time points after 4-OHT treatment to confirm the knockout of reelin (see Figure 2A). We observed a remarkable decrease in reelin expression after 2 days of treatment with 4-OHT, and complete elimination of reelin immunostaining after 5 days (Figure 2A). Using Ca^2+^ imaging, we compared spontaneous activity from wt and Reln^cKO^ OTCs. Both groups of OTCs were treated with 4-OHT. Previously, it was verified that 4-OHT treatment does not alter basic synaptic transmission [59]. To test whether reelin influences the developmental desynchronization shift around the first postnatal week, Ca^2+^ signals were recorded from wt and Reln^cKO^ OTCs at 7 DIV. In the wt OTCs, the Ca^2+^ recording analyses revealed that 70% of cortical neurons displayed desynchronized activity, and the remaining 30% of neurons were synchronized at 7 DIV (Figure 2F). On the other hand, in recorded Reln^cKO^ OTCs, cortical neurons showed only 50% of a desynchronized network pattern, while the remaining 50% of the recorded neurons were synchronized (Figure 2F). AT 14 DIV, in the wt OTCs, the Ca^2+^ recording analyses revealed that 90% of cortical neurons displayed desynchronized activity, and the remaining 10% of neurons were synchronized at 14 DIV (Figure 2B,C,G; see also Appendix A). On the other hand, in recorded Reln^cKO^ OTCs, cortical neurons showed only 10% of a desynchronized network pattern, while the remaining 90% of the recorded neurons were synchronized (Figure 2D,E,G; see also Appendix A). To test whether reelin deficiency caused a delay or complete blockade of network developmental desynchronization, we kept the OTCs until 25 DIV and performed Ca^2+^ recording in both wt and Reln^cKO^ OTCs (Figure 2H). Similar to the analyses at 14 DIV, we observed the same effect at 25 DIV, indicating that reelin deficiency caused a blockade rather than a delay in the developmental desynchronization shift (Figure 2H). This finding suggests that reelin is not only necessary for the desynchronization of the developmental network at the end of the second postnatal week but also two weeks later. However, we cannot conclude that this blockade occurs permanently because we did not record OTCs at the adult stage.

### 3.3. Wildtype Reelin Rescues Developmental Desynchronization in Reln^cKO^ Neurons

Next, we investigated whether secreted reelin from wt OTCs could rescue the abnormal synchronized neuronal activity pattern observed in the Reln^cKO^ OTCs at 14 DIV. To address this question, we co-cultured OTCs in three different conditions: wt with wt, Reln^cKO^ with Reln^cKO^, or wt with Reln^cKO^ (Figure 3A–C). Our previous research has shown that the amount of secreted reelin in the culture medium varies depending on the conditions. In the wt + wt co-culture, the amount of secreted reelin was abundant. In the wt + Reln^cKO^ co-culture, a lesser amount was detectable. In contrast, reelin was almost undetectable in the Reln^cKO^ + Reln^cKO^ co-culture medium [59]. Ca^2+^ recording experiments at 14 DIV revealed that 6% of wt + wt co-culture neurons displayed a synchronous activity pattern, which was not significantly different from the wt + Reln^cKO^ co-culture neurons (14%; Figure 3D). In contrast, a significantly higher percentage of Reln^cKO^ + Reln^cKO^ co-culture neurons exhibited a synchronous activity pattern (97%; Figure 3D). However, desynchronized activity was detected in the majority of wt + wt (94%) and wt + Reln^cKO^ (86%) co-cultures. In contrast, the fraction of desynchronized neurons in Reln^cKO^ + Reln^cKO^ co-culture neurons was remarkably reduced (3%; Figure 3D). These results suggest that reelin secreted from wt OTCs into the incubation medium enabled developmental synchronization shift in the co-cultured Reln^cKO^ neurons to a basic level comparable to the control group.

### 3.4. Developmental Network Desynchronization Is Controlled by Reelin Signaling

We further aimed to test whether the shift in developmental desynchronization defect observed in the Reln^cKO^ neurons is modulated by canonical reelin signaling. To address this question, we treated OTCs with RAP, an LDL receptor family chaperone that prevents the binding of reelin to VLDLR and ApoER2. Reelin regulates DAB1 tyrosine phosphorylation by binding to VLDLR and ApoER2 [45]. To test whether reelin signaling is involved in the developmental desynchronization shift, we performed Ca^2+^ imaging in wt and Reln^cKO^ OTCs and recorded neurons before and after bath application of RAP (50 ng/mL). At 14 DIV, wt OTCs displayed a desynchronized neuronal activity pattern; however, the bath application of RAP induced neuronal synchronization (Figure 4C; see also Appendix A). In contrast to wt OTCs, neither the proportion of synchronized nor desynchronized neurons was affected by the application of RAP to Reln^cKO^ OTCs (Figure 4C). These results suggest that blocking endogenous reelin signaling with RAP at the end of the second week results in a developmental desynchronization defect for cortical neurons.

### 3.5. GABAergic, but Not Glutamatergic, Signaling Is Involved in Reelin-Mediated Developmental Desynchronization

As previously stated, early development of SSA relies primarily on the activation of glutamatergic AMPA and NMDA receptors, and is also influenced by cholinergic activity [2,5,6]. To confirm this, we recorded Ca^2+^ signals from wt OTCs at 5 DIV when SSA is strongly dependent on AMPAR and NMDAR activation. In fact, we can see that, at 5 DIV, the bath application of either APV or CNQX can both block SSA during that developmental time window (Figure 5A,B). Additionally, reelin has been shown to enhance NMDA and AMPA receptor activity in the adult hippocampus [68]. However, during the developmental desynchronization period (around the end of the second week), it is unclear whether glutamatergic activity controls the developmental shift. To test this, we performed Ca^2+^ imaging in wt OTCs, and applied the NMDAR antagonist APV acutely at 14 DIV (Figure 5C). The results showed that bath application with APV did not affect the developmental desynchronization shift (Figure 5C). Similarly, blockade of the AMPA/kainate glutamate receptor with CNQX at 14 DIV did not affect the developmental desynchronization shift (Figure 5D). These results suggest that reelin-mediated developmental desynchronization is independent of glutamatergic signaling.

The shift in network desynchronization coincides with the developmental shift from excitatory to inhibitory GABA, which occurs at the end of the second postnatal week. Previous research has demonstrated that an increase in GABAergic inhibition may lead to the elimination of SSA in the neocortex and hippocampus [69]. To test whether the inhibitory action of GABA receptors plays a role in reelin-mediated developmental desynchronization around the second postnatal week, Ca^2+^ signals were recorded from wt OTC before and after acute bath application of specific GABA receptor antagonists. At 14 DIV, wt and Reln^cKO^ OTCs were recorded before and after acute treatment with GABA_A_R or GABA_B_R antagonists (Figure 6). Acute blockade of GABA_A_Rs with Gabazine in wt OTCs significantly reduced the fraction of desynchronized neurons in the recorded OTCs (95% for non-treated OTCs vs. 5% in Gabazine-treated OTCs; Figure 6A–C; see also Appendix A), indicating the role of the GABA_A_Rs in promoting neuronal desynchronization. Furthermore, acute blockade of GABA_A_Rs with Gabazine in Reln^cKO^ OTCs resulted in the same fraction of desynchronized neurons as in non-treated Reln^cKO^ OTCs (Figure 6C). Furthermore, acute blockade of GABA_B_Rs with CGP in wt OTCs remarkably reduced the fraction of desynchronized neurons in the recorded OTCs (93% for non-treated OTCs vs. 7% in CGP-treated OTCs; Figure 6D), whereas cute blockade of GABA_B_Rs with CGP in Reln^cKO^ OTCs resulted in the same fraction of desynchronized neurons as in non-treated Reln^cKO^ OTCs (Figure 6D), confirming the dysfunctionality of GABA_B_Rs in the Reln^cKO^ OTCs. These results suggest that both GABA_A_Rs and GABA_B_Rs are required for reelin-mediated developmental desynchronization.

## 4. Discussion 

In this study, we identified a new function of reelin in early postnatal cortical neural network development. Our findings indicate that postnatally induced reelin deficiency caused a disruption of the developmental desynchronization shift naturally occurring in the somatosensory cortical neurons around the second postnatal week. To mimic this effect, we blocked reelin binding to its receptors ApoER2 and VLDLR using RAP. As there are no specific selective antagonists for ApoER2 and VLDLR, we used RAP to broadly block LRP family members. Co-culture of Reln^cKO^ OTCs with reelin-secreting wt OTCs rescued the deficient network developmental shift from synchronized to desynchronized activity in co-cultured Reln^cKO^ OTCs. Reelin is secreted by Cajal–Retzius (CR) cells in layer I of the neocortex during early embryonic stages. The number of CR cells in layer I significantly drops after birth, and CR cells vanish completely around P14 due to selective cell death through apoptosis [62]. Therefore, after birth, reelin can be secreted from CR cells (at least until P14) but also by the inhibitory interneurons [66]. Therefore, secreted reelin from both cell types can be responsible for triggering the developmental desynchronization shift in the somatosensory cortex. Furthermore, there is another group of neurons known as ocean cells or stellate cells that express reelin exclusively in the medial entorhinal cortex. These cells have been demonstrated as being responsible for context-specific fear memory [67]. Reelin function in early postnatal development is still not well documented. For example, it was found that the N-terminal region of reelin restricts dendritic growth of neocortical apical pyramidal neurons [52]. Another study conducted a morphometric analysis of interneurons in the adult reeler neocortex and hippocampus and showed that reeler interneurons had hypertrophic growth with longer dendritic branches compared to wild-type interneurons [51]. In early postnatal development, the conditional knockout of reelin increased dendritic growth in inhibitory interneurons due to the absence of GABA_B_R signaling [53,59]. Additionally, reelin plays a role in developing vertical columnar structures in the mouse presubicular cortex. It acts as a stop signal for the growth and branching of postnatal pyramidal cell apical dendrites [70]. On the other hand, cortical neurons’ apical dendrites are inhibited by chondroitin sulfate proteoglycans (CSPGs), which are highly expressed in the marginal zone, and a recent study has shown that reelin signaling in the marginal zone counteracts the inhibitory action of CSPGs on apical dendrites in the cortex [71]. Taken together, this study contributes to our comprehension of reelin’s function during early postnatal cortical development.

The SSA depends on glutamatergic signaling in the early development of the neocortex [2,5,6,56,72,73,74]. Furthermore, it has been proposed that the maturation of GABAergic signaling is responsible for the developmental desynchronization shift in the neocortex and hippocampus [2,4,69]. The present study also aimed at investigating the potential involvement of glutamatergic signaling in the developmental desynchronization shift during the second postnatal week of development. However, the blockade of glutamate receptors did not affect the developmental shift to desynchronization, indicating that glutamatergic signaling is limited to the first postnatal week when SSA is the predominant cortical activity. While the pattern of SSA during the first postnatal week is well-studied, there is insufficient information on how the developmental desynchronization shift occurs during the second postnatal week. Several studies have found that neocortical networks undergo a transition to a desynchronized state between the end of the second and third postnatal weeks of development [7,20,21,22,23]. However, the underlying mechanism of this developmental desynchronization shift remains largely unexplored. Recently, it was shown that GABA_A_Rs, but not GABA_B_Rs, modulate network desynchronization in the medial entorhinal cortex (mEC) during the second postnatal week of development [75]. Our results confirm that GABA_A_Rs are involved in the developmental desynchronization shift in the somatosensory cortex. During early development, GABA_A_Rs do not mediate hyperpolarization-dependent inhibition, since GABA_A_R signaling is predominantly depolarizing and excitatory [76]. The switch from GABA excitation to inhibition occurs in the neocortex between the first and the second postnatal developmental week [27,77]. There is a link between reelin and postsynaptic GABA_A_R in the grid cells of the entorhinal cortex. Reelin-positive cells selectively express the α3 subunit of the GABA_A_R and this selective pattern is responsible for both tonic and phasic inhibition [78]. Furthermore, the neurohormone oxytocin desynchronized SSA in the first postnatal week in the mouse medial prefrontal cortex. By the end of the second postnatal week, the inhibition of GABA_A_Rs restored SSA, suggesting the emergence of GABA_A_R-mediated inhibition as a major factor in the termination of SSA in the mouse medial prefrontal cortex [23]. Moreover, reelin haploinsufficiency in the mouse neocortex disrupts the developmental GABA excitation/inhibition switch, indicating an interaction between reelin signaling and GABA receptors. It is important to note that the postnatal GABA shift is not a simple switch that takes place at a certain moment in postnatal development, but rather reflects a gradual change in neuronal Cl homeostasis, such that GABA signaling gradually becomes more hyperpolarizing within local neuronal networks [27]. The timing of the GABA excitation/inhibition switch coincides with the period when neocortical activity becomes desynchronized. The transition of the network from SSA to an independent desynchronized state may be determined by the GABA excitation/inhibition switch.

GABA_B_Rs are metabotropic receptors coupled to guanine nucleotide-binding proteins (G proteins). They modulate Ca^2+^ and K^+^ channels, inducing both presynaptic and slow postsynaptic inhibition [79]. The presynaptic GABA_B_Rs are coupled to Ca^2+^ channels, which regulate the release of neurotransmitters [80]. During prenatal development in vivo, GABARs were detected by immunocytochemistry as early as E14 in the marginal zone and the subplate of the cortex, indicating the expression and potential formation of functional GABARs [81]. During embryonic development, GABA is the main neurotransmitter. At this stage, GABA is not hyperpolarizing because GABAR lacks coupling between G proteins and potassium channels until the end of the first postnatal week [82]. The non-hyperpolarizing activation of GABAR in early embryonic development has been shown to promote dendritic growth [83]. However, after birth, when GABA_B_R shifts to hyperpolarization, it can couple to G proteins and potassium channels. Previous studies have shown that GABA_B_R intracellular signaling via Gαi/o proteins is impaired in Reln^cKO^ mice [59]. Therefore, in the absence of early postnatal reelin expression, the GABA_B_Rs were not functional [59]. Our findings suggest that the activation of GABA_B_Rs in the somatosensory cortex is necessary for reelin-mediated developmental desynchronization shift. 

## 5. Conclusions

This study documents the role of an extracellular matrix protein in the developmental desynchronization shift that occurs naturally at the end of the second postnatal week of cortical development. In the absence of reelin neuronal activity, Reln^cKO^ OTCs exhibited synchronous neuronal activity at the end of the second postnatal week. Furthermore, we demonstrated that developmental desynchronization impairment observed in Reln^cKO^ could be rescued when Reln^cKO^ OTCs were co-cultured with wt OTCs. Finally, our findings indicate that the developmental transition to a desynchronized state at the end of the second postnatal week is not dependent on glutamatergic signaling. Rather, the transition is dependent on GABAergic signaling. The results indicate that reelin controls developmental desynchronization through GABAergic signaling. These results are consistent with a previous study which demonstrated that GABARs modulate network desynchronization in the mEC during the second postnatal week of development [75]. The results of the present study confirm that GABARs are involved in the developmental desynchronization shift in the somatosensory cortex. However, it shows that this process is dependent on the upstream reelin signaling pathway. The present study suggests that reelin also plays a role in shaping neuronal activity during early postnatal network activity development.

## Figures and Tables

**Figure 1 biomolecules-14-00593-f001:**
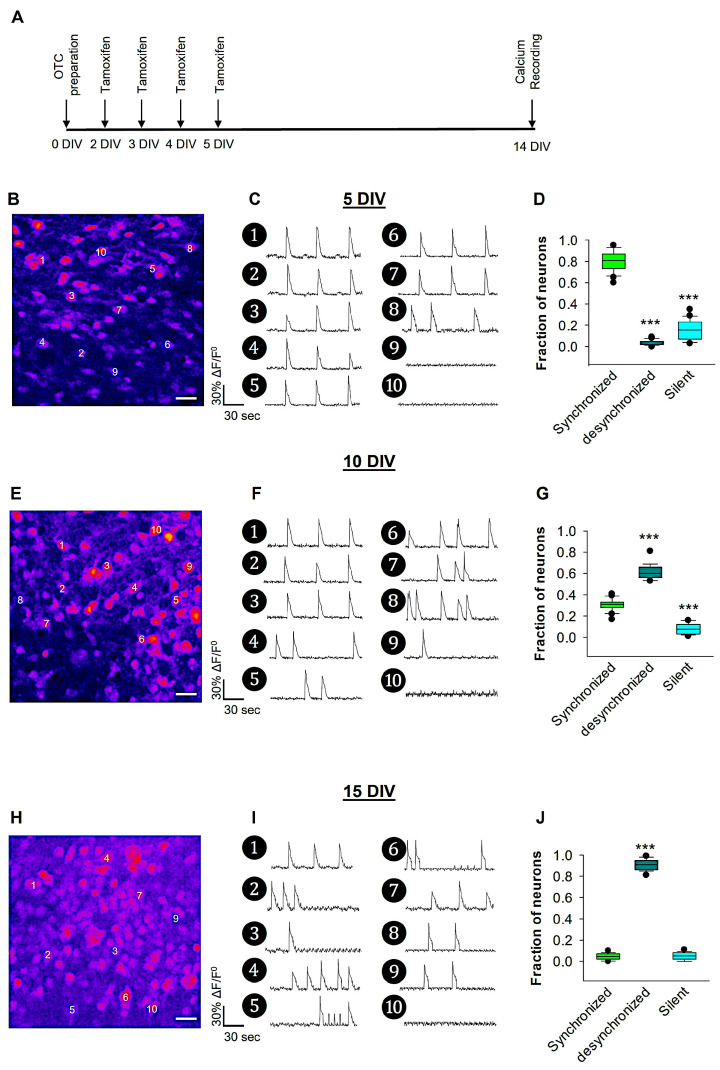
Recording of neuronal activity in wt at different developmental stages. (**A**) Graphic flow chart of experimental procedures. For this experiment, OTCs were prepared from 8 wt and 8 Reln^cKO^ mice. An example image during Ca^2+^ recording of an OTC loaded with OGB-1 at 5 DIV is shown in (**B**). The ΔF/F^0^ traces of 6 cells (numbered in B) are displayed in (**C**), which show mostly synchronized, and less frequent desynchronized and silent cells. The fraction of neurons participating in synchronized, desynchronized, and silent neuronal activity is represented in the box plot in the graph (**D**). A total of 25 OTCs were used for the analyses at 5 DIV. An example image during Ca^2+^ recording of an OTC loaded with OGB-1 at 10 DIV is shown in (**E**). The ΔF/F^0^ traces of 10 cells (numbered in E) are displayed in (**F**), which show mostly desynchronized, less frequently synchronized, and few silent cells. The fraction of neurons participating in synchronized, desynchronized, and silent neuronal activity is represented in the box plot in the graph (**G**). A total 28 OTCs were used for the analyses at 10 DIV. An example image during Ca^2+^ recording of an OTC loaded with OGB-1 at 14 DIV is shown in (**H**). The ΔF/F^0^ traces of 6 cells (numbered in H) are displayed in (**I**), which show predominantly desynchronized and few synchronized silent cells. The fraction of neurons participating in synchronized, desynchronized, and silent neuronal activity is represented in the box plot in the graph (**J**). A total of 27 OTCs were used for the analyses at 14 DIV. Scale bars in B, E, and H: 30 μm. One-way-ANOVA on Ranks followed by Tukey’s multiple comparison test; *** *p* < 0.001. The horizontal lines in the box plots represent the median; whiskers indicate variability outside the upper and lower quartiles; box indicates the middle half of the sample.

**Figure 2 biomolecules-14-00593-f002:**
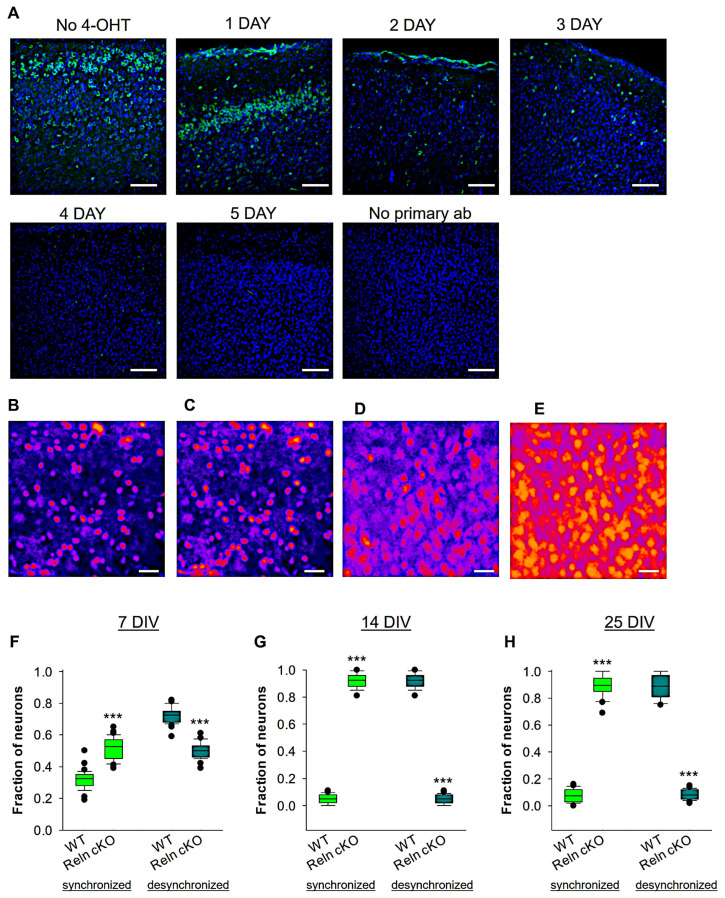
Comparison of neuronal activity between wt and Reln^cKO^ mice. Confocal merge images of reelin immunofluorescence (in green) and the nuclear marker TO-PRO-3^®^ (in blue) were obtained after 4-OHT treatment for 1–5 days starting from 2 DIV. (**A**) Both positive control (No 4-OHT) and negative control (no primary antibody against reelin) are shown. Scale bars in A are 50 μm. For Ca^2+^ recording experiment, OTCs were prepared from 11 wt and 12 Reln^cKO^ mice. An example of images taken during Ca^2+^ recording of a wt OTC loaded with OGB-1 at 14 DIV is shown in (**B**,**C**) at different time intervals during the recording (see also Appendix A). Note that at 14 DIV the dominant type of activity in wt OTCs is the desynchronized activity. Example images during Ca^2+^ recording of Reln^cKO^ OTC loaded with OGB-1 at 14 DIV in Reln^cKO^ OTCs are shown at different time interval during the recording (**D**,**E**; see also Appendix A). Note that at 14 DIV, the primary type of activity in Reln^cKO^ OTCs is the synchronized activity. Scale bars in B-E are 30 μm. The box plot in the graph represents the fraction of neurons participating in synchronized and desynchronized neuronal activity at 7 DIV (**F**; 21 wt OTCs and 20 Reln^cKO^ OTCs), 14 DIV (**G**; 24 wt OTCs and 23 Reln^cKO^ OTCs), and 25 DIV (**H**; 18 wt OTCs and 17 Reln^cKO^ OTCs). The Mann-Whitney U test was used for statistical analysis and the results showed *** *p* < 0.001. The median is represented by the horizontal lines in the box plots, while the whiskers indicate variability outside the upper and lower quartiles. The box indicates the middle half of the sample.

**Figure 3 biomolecules-14-00593-f003:**
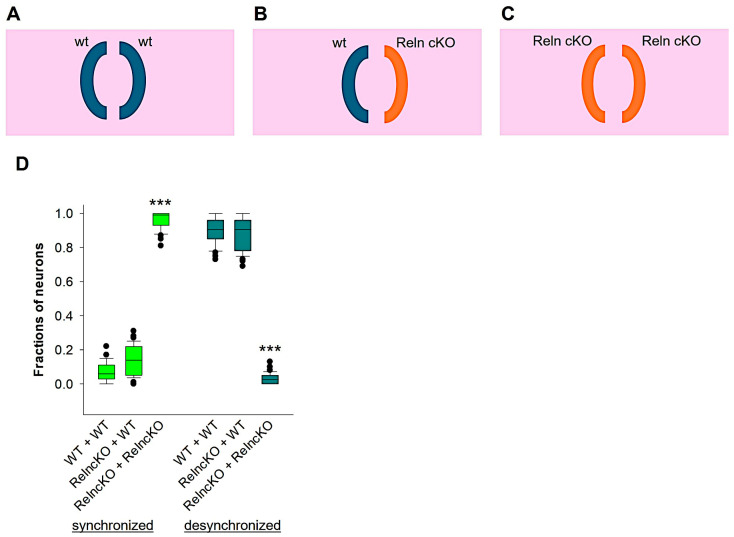
Neuronal desynchronization in Reln^cKO^ OTCs is rescued by secreted reelin. Schematic representation of the co-culture experimental approach. (**A**): Two co-cultured wt OTCs. (**B**): A wt OTC co-cultured together with a Reln^cKO^ OTC. (**C**): Two co-cultured Reln^cKO^ OTCs. The fraction of neurons participating in synchronized and desynchronized neuronal activity is represented in the box plot in the graph (**D**). Statistical analysis was performed using one-way ANOVA on ranks followed by Tukey’s multiple comparison test, with a significance level of *** *p* < 0.001. For this analysis, we used a total of 32 OTCs from wt and 36 OTCs from Reln^cKO^ mice. Note that the fraction of synchronized and desynchronized neurons recorded from Reln^cKO^, which were co-cultured with wt OTCs, showed no difference from the wt OTCs that were co-cultured with wt OTCs. The Mann–Whitney U test was used for statistical analysis, and the results showed *** *p* < 0.001.

**Figure 4 biomolecules-14-00593-f004:**
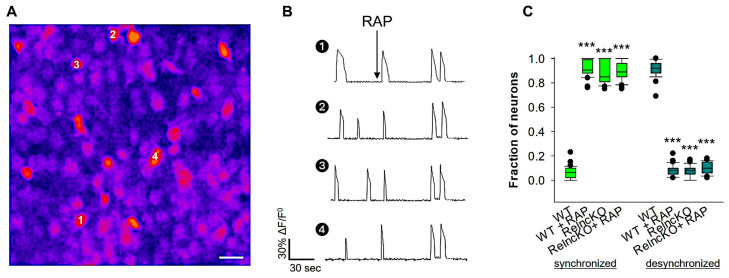
Effect of RAP blockade of reelin receptors on network desynchronization. Wt and Reln^cKO^ OTCs were recorded before and after bath application of RAP (50 µg). For each experiment, OTCs were prepared from 7 wt mice and 7 Reln^cKO^ mice. Control wt OTCs were treated with H_2_O. A sample image of an OGB-1-loaded OTC used for Ca^2+^ recording at 14 DIV is shown in (**A**). The ΔF/F^0^ traces of 4 cells (numbered in A) are shown in (**B**), demonstrating that RAP application to wt OTCs dramatically reduced the fraction of desynchronized OTC after RAP treatment. The fraction of neurons involved in synchronized and desynchronized neuronal activity is shown in the box plot in graph (**C**). In contrast to wt OTCs, neither the proportion of synchronized nor desynchronized neurons was affected by the application of RAP to Reln^cKO^ OTCs. A total of 42 OTCs were used for the analyses. The Mann–Whitney U test was used for statistical analysis, and the results showed *** *p* < 0.001.

**Figure 5 biomolecules-14-00593-f005:**
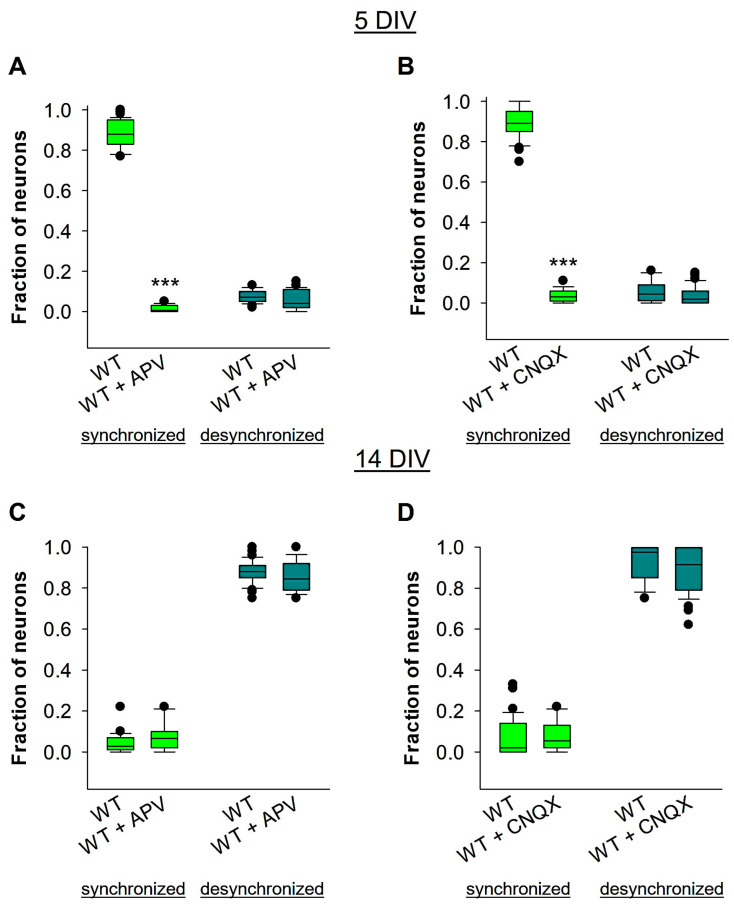
Reelin-mediated developmental desynchronization is independent of glutamatergic signaling. OTCs from wt mice were recorded before and after bath application of either APV or CNQX. In each experiment, OTCs were prepared from 8 wt mice. Control wt OTCs were treated with H_2_O. The fractions of neurons involved in synchronized and desynchronized neuronal activity before and after treatment with APV at 5 DIV are presented in (**A**), and at 14 DIV are presented in (**C**). The fractions of neurons participating in synchronized and desynchronized neuronal activity before and after treatment with CNQX at 5 DIV are presented in (**B**), and at 14 DIV are presented in (**D**). A total of 52 OTCs were used for analyses of the APV experiments, and 48 OTCs were used for analyses of the CNQX experiments. The Mann–Whitney U test was used for statistical analysis, and the results showed *** *p* < 0.001.

**Figure 6 biomolecules-14-00593-f006:**
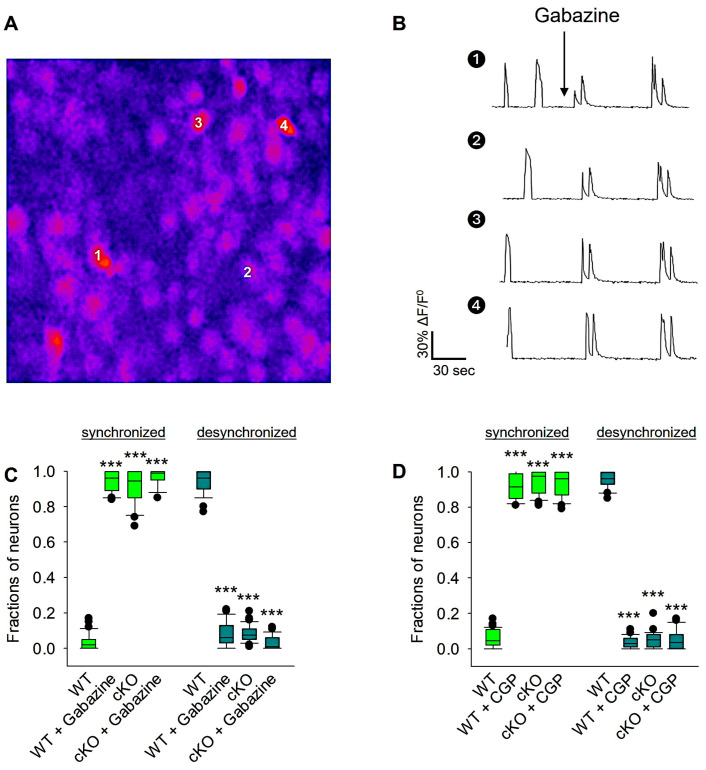
Reelin-mediated developmental desynchronization is dependent on GABAergic signaling. Reln^cKO^ or wt OTCs were recorded before and after bath application of either Gabazine or CGP. In each experiment, OTCs were prepared from 12 wt and 12 Reln^cKO^ mice. Control wt or Reln^cKO^ OTCs were treated with H_2_O. An example image during Ca^2+^ recording of an OTC loaded with OGB-1 at 14 DIV is shown in (**A**). An example of ΔF/F^0^ traces of 4 cells (numbered in A) before and after Gabazine treatment is shown in (**B**). The fraction of neurons involved in synchronized and desynchronized neuronal activity before and after treatment with Gabazine is shown in the box plot in the graph (**C**). The fraction of neurons participating in synchronized and desynchronized neuronal activity before and after treatment with CGP is shown in the box plot in the graph (**D**). A total of 36 OTCs were used for analyses of the Gabazine experiments, and 35 OTCs were used for analyses of the CGP experiments. Statistical analysis: one-way ANOVA on ranks followed by Tukey’s multiple comparison test; *** *p* < 0.001.

## Data Availability

The Relnflox mice (B6.129-Relntm1Her/J; Stock # 037348) are available from Jackson Laboratories, Bar Harbor, Maine.

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
