# Peer review of "Reelin Regulates Developmental Desynchronization Transition of Neocortical Network Activity"

_biomolecules, 2024, doi:10.3390/biom14050593_

Round 1
Reviewer 1 Report
Comments and Suggestions for Authors
Hamad et al investigated the effect of Reelin on the transition from spontaneous synchronous activity to a desynchronized firing pattern in early postnatal neocortical neurons. They measured Ca++ signals in organotypic brain slices and showed that removal of Reelin signaling (conditional postnatal KO of Reelin or incubation with RAP for the inhibition of Reelin receptors) prevented this transition that occurs at the end of the second postnatal week. This absence of desynchronization can be rescued when the Reelin-deficient slice is co-cultured in the presence of a wild-type slice. Finally, they used the NMDA receptor agonist APV and AMPA/kainite glutamate receptor inhibitor CNQX on their slice cultures to show that the effect of Reelin is not dependent on glutamatergic signaling. However, the use of GABA receptors antagonists suggests the involvement of GABAARs and GABABRs.
This is a nice work revealing an interesting new function for the Reelin pathway.
Here are some suggestions that could be used to improve the manuscript:
Fig 2:
-Is it a delay in desynchronization or a complete inhibition in the Reelin-deficient slices? Could you keep the slices longer and test Ca++ later?
-The slices were treated 4 times with 4-OTH, starting from DIV 2, to knock-out reelin in slice cultures. Please provide a western blot to determine at what time during this procedure Reelin becomes absent within the slices.
-It seems from the time lapses that the observed phenotype in the absence of Reelin is more than just being synchronize at 14 DIV. It seems that the signal is more frequent and stronger. Do you observe that difference when comparing the wild-type and cKO slices at earlier time point when the WT is still synchronized (for instance at 4 or 7DIV WT vs cKO)?
Fig4:
-As you did with the GABA receptor antagonists, a nice control to make sure that the effect of RAP is due to the inhibition of the Reelin pathway is to use it on cKO slices.
Reviewer 2 Report
Comments and Suggestions for Authors
It is an interesting work providing data on the possible role of reelin in postnatal cortical developmental desynchronization using a conditional reelin knock-out (RelncKO) mouse model. Some trivial mistakes (see following specific points) affect the manuscript and the clarity of the description.
“Conclusions” section is rather speculative and is not focusing on the main results regarding the demonstration of a reelin-mediated developmental desynchronization.
Specific points
Please revise the legend of Fig. 1 page 5 lines 170-188: 6 cells or ten cells??? Lines 185: “A total of 27 OTCs were used for the analyses at 10 DIV”: or 14 DIV????
Please revise carefully the legend of Fig. 2 page 7 and 8: line 287: “Note that at 14 DIV the dominant type of activity in wt OTCs is the synchronized activity” this is in contrast with what is shown in figure 2 and described in the result section. Line 290: “Note that at 14 DIV, the primary type of activity in RelncKO OTCs is the desynchronized activity”: this is in contrast with what is shown in figure 2 and described in the result section.
In section 3.5 at line 358 note the following statement “To test this, we performed Ca2+ imaging in wt and RelncKO OTCs, and applied the NMDAR antagonist APV acutely (Figure 5A)”. Indeed, results shown in Fig. 5A are not corresponding to the previous statement: synchronized or desynchronized status are only related to developmental stage in WT OTCs.
…applied the NMDAR antagonist APV acutely… Any dose-response performed? Any functional test of real blockade of NMDA or AMPA receptors in the tested conditions?
Pag. 11 line 387: “Furthermore, acute blockade of GABAARs with CGP in RelncKO OTCs resulted in the same fraction of desynchronized neurons as in non-treated RelncKO OTCs (Figure 6C)”. In this sentence CGP should be replaced by Gabazine.
Pag. 12 line 397 legend of Figure 6. “Reelin-mediated developmental desynchronization is dependent on glutamatergic signaling” should be “Figure 6. Reelin-mediated developmental desynchronization is dependent on GABAergic signaling”.
Are GABAAR or GABABR antagonists affecting reelin secretion in OTCs Since RAP experiments suggest a very rapid time-course for the effect of reelin on network desynchronization, this possibility cannot be excluded. Moreover, GABAergic interneurons are mutually coupled. As reported by the authors, after birth, the dominant cell type that expresses reelin in the cortex are the inhibitory GABAergic interneurons.
Pag 14 line 499: “Our findings are consistent with previous research that demonstrates the termination of the UP state is mediated by the activation of GABABRs [82]”: an explanation of UP state is necessary.
Pag. 13 line 422-424: “The disappearance of CR cells coincides with the emergence of desynchronized cortical networks in the cortex. This suggests that CR-reelin secreting cells may play a key role in refining the developmental shift towards desynchronization”. The connection of this well-established observation with the reported results (showing that reelin is mediating desynchronization) is not well explained. The following sentence is not adequately explained at the light of the reported results (“After birth, the dominant cell type that expresses reelin in the cortex are the inhibitory GABAergic interneurons).
Round 2
Reviewer 1 Report
Comments and Suggestions for Authors
I am satisfied by all the responses but one. The authors did not comment on my observation from the time lapse about a more frequent and stronger signal in the absence of Reelin. I asked whether they would also observe this at an earlier stage when desynchronization is not yet (or barely) apparent.
I confirm here that the paper should be published but I would appreciate if I could get an answer to this question.
Reviewer 2 Report
Comments and Suggestions for Authors
Authors have adequately addressed all points in report 1
Author Response
We would like to express our gratitude once more to the reviewer for his insightful comments.